# Investigating the Environmental Benefits of Novel Films for the Packaging of Fresh Tomatoes Enriched with Antimicrobial and Antioxidant Compounds through Life Cycle Assessment

Christina Tsouti [1,*], Christina Papadaskalopoulou [1], Angeliki Konsta [1], Panagiotis Andrikopoulos [1], Margarita Panagiotopoulou [2], Sofia Papadaki [2], Christos Boukouvalas [2], Magdalini Krokida [2] and Katerina Valta [1]

1   Draxis Environmental SA, Athens Branch, 317 Mesogeion Avenue & Lokridos, 15231 Athens, Greece
2   Laboratory of Process Analysis and Design, School of Chemical Engineering, National Technical University of Athens, Zografou Campus, 9 Iroon Polytechneiou, 15780 Athens, Greece
*   Correspondence: chtsouti@draxis.gr; Tel.: +30-210-9247134

**Abstract:** Food systems account for 21–37% of total net anthropogenic greenhouse gas emissions. At the same time, in the European Union, the retail and consumption stages account for half of the total food waste produced across the entire food supply chain. For this, there is a continuous development of novel packaging materials to extend the shelf life of fresh products and thus reduce food waste produced at these stages. The aim of the present research is to examine the environmental performance of such materials enriched with antioxidant and antimicrobial compounds by considering their effect on the shelf-life extension of packed fresh tomatoes. In particular, two novel packaging films, a film with incorporated tomato leaf-stem extract and Flavomix through extrusion and a film coated with zein nanofibers containing the aforementioned bio-active compounds through electrospinning were studied for the packaging of fresh tomatoes and compared to conventional polypropylene packaging film. An antioxidant effect was recorded for both films achieving a shelf life prolongation of three days. Moreover, both films exhibited in vitro antibacterial activity against *Staphylococcus aureus* and *Escherichia coli*. In addition, antimicrobial activity was observed against yeast and molds and the total viable bacterial count in packed fresh tomatoes. The environmental benefits were evaluated using a life cycle assessment. The results indicated a decrease in the environmental impacts by 14% considering the entire food supply chain for both novel films. The environmental performance of novel film production by extrusion shows an increased impact of 31% compared to conventional film, while nanocoating shows an increased impact of 18%.

**Keywords:** life cycle assessment; novel food packaging film; antioxidant and antimicrobial compounds; shelf life; food waste; electrospinning; extrusion

## 1. Introduction

Although food constitutes a precious commodity [1], approximately one-third of the food produced for human consumption is wasted or lost before it is consumed [1,2]. The issue of Food Waste and Loss (FWL) has attracted increased international attention due to its environmental, economic, and social impacts [1,3–5], as reflected in the United Nations 2030 Agenda for Sustainable Development [3,4]. In particular, Target 12.3 of the Sustainable Development Goals (SDGs) calls for halving the global food waste per capita at the retail and consumer level by 2030 and for a reduction in food losses in production and supply chains, including post-harvest losses [2,3,6]. FWL is generated through the entire food supply chain, including primary production; post-harvest processing and storage; processing and manufacturing; packaging; distribution and retail; restaurants and food services; and households [1–3]. FWL quantification along the food supply chain is quite challenging due to existing differences between the definition of food waste and food loss as well as the

methodological framework for measurement in the various stages of the supply chain. Such a study was conducted by Caldeira et al. [7] focusing on the quantification of food waste per product group along the food supply chain in the European Union (EU) in 2011. In the study by Caldeira et al. [7], food waste was defined as the fractions of 'food and inedible parts of food removed from the food supply chain' to be recovered or disposed' [7,8]. Based on the outcomes of the study, the largest share (46% of the total food waste) is produced during consumption, followed by primary production (25%), processing (24%), and finally the distribution and retail stage (5%), while the largest amount of food waste was estimated for fruit and vegetables [7]. Given the above, the retail and consumption stages account for half of the total food waste produced across the whole Food Supply Chain (FSC). This phenomenon is not easy to interpret since several parameters influence the amount of food wasted within these two stages, such as consumer behavior, product handling, packaging materials and conditions, marketing approaches, etc. In addition to this, what is even more challenging is the identification of the interlinkages between these parameters since there is neither a clear nor a linear cause-effect relationship between them.

As mentioned, packaging is one of the various parameters affecting the amount of FWL produced. The global food packaging market was estimated at USD 300 billion in 2019 and is expected to grow by USD 60 billion by 2025 [9]. The role of packaging is to protect a product from factors that could cause its degradation and maintain its quality to acceptable levels. Especially for fresh products, the inactivation of remaining microorganisms that could cause quality deterioration is very essential. Food spoilage occurs when food is characterized as unsuitable for consumption as a result of biochemical changes caused by the spread of microbial populations in it [10]. Conventional packaging acts as a passive barrier which isolates the product from its environment, protecting it from external factors (e.g., heat, oxygen, moisture, insects), while providing a safe and stable means of transportation [11]. Polymeric films are among the most common and preferable materials adopted for this function as they are flexible; versatile; easily adaptable in terms of thickness, shape, or color; and can be generated with many methods (extrusion, solvent casting, electrospinning) to meet the needs of each product [12]. This type of packaging contributes to moisture control and reduces contamination during handling. However, consumers today demand more natural, less processed food products without synthetic preservatives [13]. For this reason, there is a continuous development of new packaging technologies to enhance the stability, quality, and sustainability of food products, extending their shelf life without the need for chemical additives [14,15]. One such promising technique is the Modified Atmosphere Packaging (MAP), usually applied to extend the shelf life of various types of food [16]. Through MAP, the atmosphere inside a package is either completely removed or altered by changing the composition of the gases inside the package, which results in a reduction in food spoilage [17]. Another innovative approach is packaging using active materials. Based on the EU regulations 1935/2004/EC and 450/2009/EC, "active materials refer to materials that are intended to extend the shelf life or to maintain or improve the condition of packaged food" [18]. In this advanced technology, active agents (essential oils, plant extracts, carbon/oxygen absorbers, etc.) are embedded in the commercial polymeric matrices and interact with the food and/or the environment, sustaining and prolonging the shelf life [19]. Moreover, there is increased interest in the use of nanomaterials in food packaging. Nanotechnology has revealed new opportunities for the creation of innovative systems for the delivery of active compounds [14,20]. In this context, the investigation of new active packaging technologies such as antimicrobial and/or antioxidant-loaded nanostructures and modified conventional materials [11,20–23] for fruits, vegetables, meat, and bread appears very attractive to researchers worldwide.

As it was highlighted before, packaging is essential for preserving and delivering food to the consumers in its best condition. Nevertheless, apart from ensuring both the safety and the integrity of food, current advances in packaging provide additional services across the food supply chain [24]. Among these services, the contribution of packaging in products'

shelf life extension, as well as in the reduction in the food waste generated, has drawn the attention of both the scientific and the business world. Regulation (EC) No. 2073/2005 on microbiological criteria for foodstuffs and specifically Article 2(f) provides the definition of 'shelf-life', stating that it means either the period corresponding to the period preceding the 'use by' or the minimum durability date, as defined, respectively, in Articles 9 and 10 of Directive 2000/13/EC. Directive 2000/13/EC was repealed and replaced by Regulation (EU) No. 1169/2011 on the provision of food information to consumers. In Regulation (EU) No. 1169/2011 and specifically in Article 2(r) the "date of minimum durability of a food" is defined as the date until which the food retains its specific properties when properly stored [25].

Given the fact that packaging can contribute to shelf-life extension and that it comes after the primary production operations and handling, attention is paid to its contribution in reducing food waste at the retail and consumption stages of the food supply chain. However, due to the lack of data, a comprehensive approach for quantifying the relationship between shelf life and food waste reduction is not yet identified [13,26]. As Amani et al. [27] mention, a straightforward relationship between shelf life and food waste is not evident, as several parameters such as complex consumption patterns, long food supply chains, and increased demand for "fresh products" need to be taken into consideration [27].

Research on the use of food waste and byproducts such as peels, pomace, and seeds has increased recently. Bioactive components, which are highly concentrated in waste and byproducts from the primary sector and food manufacturing industries, offer opportunities for the food packaging sector. Among these opportunities are the development of biodegradable polymers, functional chemicals, and active packaging materials. The Food and Agriculture Organization of the UN estimates that 413 MT of food is wasted during agricultural production, followed by the postharvest stage (293 MT) and food processing (148 MT) [28]. Given the seriousness of the situation, the European Union adopted SDG 12.3, which seeks to "halve per capita food waste at the retail and consumer levels and reduce food losses along production and supply chains (including post-harvest losses) by 2030." Under this frame, the valorization of waste and by-products generated from the fruit and vegetable processing industry is promising due to their significant high amounts of bioactive compounds, such as antioxidants, fiber, and antimicrobials, posing them as the most suitable candidates for the development of active packaging. There are a lot of recent studies that deal with the use of agri-food waste for this purpose [29]. Specifically, naturally derived from food waste (citrus pomace, peels and seeds, grape pomace, olive pomace, olive tree leaves, apple peels, peach and apricot stones and peels, tomato stem, leaves, peels and seeds, etc.), antioxidants, and/or antimicrobial compounds such as phenolics, flavonoids, polyphenols, terpenes, acids (ascorbic, citric, etc.), and pigments (curcumin, carotenoids, chlorophyll, quercetin, anthocyanin, etc.) [30–37] are considered as effective compounds in the development of active food packaging.

Recently, it has not only been reported that food systems have a significant contribution on environmental pressures globally, accounting for 21–37% of total net anthropogenic greenhouse gas (GHG) emissions [38], but also it has been highlighted that these pressures call for coordinated action by all parties involved across the food supply chain, including consumers [39]. FSC can be quite dynamic, complex, and heterogenous, while a variety of stakeholders operate within its boundaries. Given the aforementioned points and taking into consideration that agri-food products can travel through several stages and pathways during their life cycle, the environmental impacts of different stages across the FSC should be examined when it comes to the full accounting of food systems [40].

The life cycle assessment (LCA) method has been constantly gaining ground both in accounting and interpreting the environmental performance of various agri-food products and FSC per se, as well as in examining the effects of technological, managerial, and behavioral interventions across different stages within the food system. An extensive literature review of LCA studies on food systems, carried out by Poore et al. [39], highlighted the fact that significant gaps and methodological issues arise among the various studies ex-

amined [39]. Although it is well recognized that the agricultural production accounts for the majority of the environmental impacts of the food system, Molina et al. [41] question whether the food LCAs can deliver the full picture, when it comes to encapsulating direct and indirect impacts arising in other stages of the FSC and specifically about the effect of packaging material on the products' life cycle [41]. Although there are several LCA studies about the environmental effect of applying different packaging materials on various food products [13,15,42–51], significant gaps exist on how packaging can affect other stages of the FSC. As Molina et al. [41] state, when it comes to the environmental assessment of packaging, one should go a step further from just accounting for the impacts due to production and waste treatment of packaging material [41]. Regarding this matter, Pauer et al. [52] state that three additional aspects should be taken into consideration when assessing the environmental sustainability of packaging, i.e., the direct environmental impacts caused by the production and disposal of packaging; the indirect environmental impacts caused by, e.g., packaging-related FLW; and the circularity of packaging [52]. These suggestions are further reinforced by the common and harmonized methodological framework at the EU level to conduct LCA studies of plastic products from different feedstocks, proposed by the Joint Research Center (JRC) [53]. The "Plastics LCA method", developed by JRC, suggests that when comparing different packaging alternatives that lead to different amounts of food lost/wasted across distribution and/or use stages, the LCA results shall be additionally calculated including not only the loss/waste of packaging, but also the contribution of the life cycle of food loss/waste [53]. However, a common approach to integrating food loss/waste in LCA studies has not yet been developed, leading to the adoption of different approaches that can lead to non-comparable results [54].

Considering the above, there are several LCA studies that have considered the linkage between packaging material and reduction in food wasted through the perception of the extended shelf life of the product [13,15,42,44,46,50,51,55]. In particular, focusing on dairy products, Manfredi et al. [44] studied the environmental impact from the application of an antimicrobial coating in the packaging of fresh milk. Additionally, Settier-Ramirez et al. [13] examined the environmental implications of active packaging for pastry cream, while Gutierez et al. [46] studied the environmental performance of MAP packaging alternatives for cheesecake. With regard to meat products, Zhang et al. [45] investigated the environmental performance of active packaging for fresh beef, accounting for its effect on food loss reduction. Regarding fresh fruits and vegetables, Vigil et al. [15] analyzed the environmental benefits arising from active packaging for fresh cut lettuce. Finally, Adobati [42] investigated the environmental performance of active packaging solutions for fresh raspberries and strawberries. To the best of our knowledge, there is no LCA research studying the environmental performance of packed fresh tomatoes using active packaging, while also considering the entire supply chain and food waste reduction due to shelf life extension. The only study focusing on the same agri-food product (i.e., tomato) has been carried out by the USEPA [56]; however, only conventional packaging alternatives were considered, not including active packaging solutions. Furthermore, the study did not consider the entire tomato supply chain, excluding from its system boundaries food wasted at the consumption level.

Considering the above, the aim of this study is to investigate the environmental performance of fresh tomatoes packed with active packaging films, taking into account the effect of novel films on shelf life extension, the reduction in waste generation, and the reduction in transportation and waste treatment needs, across the entire food supply chain.

## 2. Materials and Methods

This section is divided into three main sub-sections. The first one refers to the laboratory tests carried out for estimating the shelf life of tomatoes achieved with the different packaging materials examined, while the second one presents the methodology for integrating shelf life into the food waste calculation. The last section presents the LCA methodology applied for this study.

### 2.1. Preparation and Characterization of Antioxidant and Antimicrobial Properties of the Packaging Films

Three different packaging materials were produced in this study. The first was a conventional polypropylene film produced by blown extrusion [57]. The second was also a blown extruded film with the difference being that Flavomix and tomato leaf-stem extract in a form of oil-based emulsion were incorporated into the last barrel of the extruder, avoiding the extensive exposure of the bioactive agents to temperatures around 160 °C and 180 °C for a longer period of time. Finally, the third was a coated PP film. More specifically, a plain PP film was used as the collector for the production of electrospun zein nanofibers. The electrospinning process was used for the encapsulation of the mixture of Flavomix and tomato leaf-stem extract in zein prolamin and the formation of an extra layer of the produced nanofibers on the conventional PP film. The extrusion technology for the incorporation of bioactive agents in plastic films is a conventional and commercially available method [58,59], while the electrospinning process has been used on a lab scale in different studies with great potential [60]. The antioxidant activity of both produced films was evaluated using DPPH (2,2-Diphenyl-1-picrylhydrazyl). The samples were prepared by mixing film slices of approx. 0.5 g with 10 mL of methanol and were stirred vigorously for 48 h at ambient conditions. The supernatant was analyzed after filtration. Regarding the antimicrobial activity, the ISO 22196:2011 (Measurement of antibacterial activity on plastics and other non-porous surfaces) [61] was applied towards *Staphylococcus aureus* (Gram) and *Escherichia coli* (Gram negative).

### 2.2. Shelf Life Measurements

In our study, fresh tomatoes were packed in three types of packaging film produced by the aforementioned methodology: (a) a plain Polypropylene (PP) film (PF—Film a), (b) a PP film with incorporated tomato leaf-stem extract and Flavomix through extrusion (EF—Film b), and (c) a PP film coated with zein nanofibers containing the aforementioned bioactive compounds through electrospinning (CF—Film c). For our experiments, three different temperatures (5 °C, room temperature (≈15 °C), and 45 °C (as an accelerated storage condition)) and a relative humidity of 80% for all of the temperature conditions were studied. The stability of the tomatoes in terms of weight loss, appearance, pH, and microbial growth was tested. To count the microflora content, the tomato samples were mixed at a dilution of 10:1 and were homogenized using a blender. The homogenates were injected in compact dry discs under aseptic conditions. The specifications and levels of acceptance for microbial growth are presented in Table S1 of the Supplementary Materials. The pH measurement was carried out for the tomato juice of the samples with a portable pH meter.

### 2.3. Integration of Shelf-Life Extension into Food Waste Calculation

In the context of the current study, the correlation between food waste reduction and shelf-life extension of packaged tomatoes due to the alternative packaging options was integrated into the LCA analysis. As was analyzed earlier, the quantification of food waste across the entire FSC is a rather complex issue, affected by several parameters. In the current study, the quantification of food wasted across the food supply chain was based on the studies of Caldeira et al. and Mena et al. [7,62], with the first focusing on the European context and the later on the United Kingdom. Based on their results, the food waste was calculated as a percentage of the total agricultural production of tomatoes, taking into consideration that no manufacturing took place, and excluding the catering sector from the consumption stage, leaving households as the only food waste producer at the consumption stage. The results regarding the distribution of tomato food waste within the major stages of the packed tomatoes value chain (agricultural production, storage, distribution, and consumption) are presented in Table 1 and were used as the baseline for calculating the FW that are avoided due to the product's shelf-life extension achieved with the alternative packaging options.

**Table 1.** FW as a percentage through the entire value chain of packed tomatoes ([7,62]).

| FSC Stage | Agricultural Production | Storage and Packaging | Distribution and Retail | Consumption |
|---|---|---|---|---|
| % FW with regard to total production | 23% | 4.5% | 1.5% | 21% |

Using the above as a point of departure and to quantify the amount of food waste avoided due to the extended shelf life of the alternative packaging scenarios, the Shelf-Life Ratio (SLR) proposed by Casson et al. [63] was adopted. In their study, Casson et al. [63] proposed an approach to encapsulate the effect of shelf-life extension due to packaging alternatives in food waste produced, taking into consideration the retail and consumption stages of the food supply chain, since the shelf life of the product is estimated after the packaging stage. The equation used to correlate the shelf-life performances of the studied alternatives is given below [63]:

$$\text{SLR} = \text{RSL/SSL} \tag{1}$$

where:

SLR—Shelf-life ratio;
RSL—Reference Shelf life (days) refers to the scenario with the worst performance concerning shelf-life;
SSL—Studied Shelf life (days) refers to the shelf life of the alternatives examined.

The respective figures for the three examined packaging films are presented in Table 2.

**Table 2.** Shelf life duration in days used in the current study for each of the examined packaging film types.

| Parameters | Origin | Packaging Film (PF—Film a) | Packaging Film (EF—Film b) | Packaging Film (CF—Film c) |
|---|---|---|---|---|
| SSL (days) | Lab experiments | 5 | 8 | 8 |
| RSL (days) | Lab experiments | 5 | 5 | 5 |
| SLR | Calculation based on Equation (1) | 1 | 0.63 | 0.63 |

Having identified the SLR, Casson et al. [63] propose an integration approach by quantifying the Potential Food Waste (PFW) produced at the retail and consumption stages of the food supply chain. The quantification is based on the following equation [63]:

$$\text{PFW} = \text{SLR} \times \text{FW} \tag{2}$$

The results obtained for each scenario regarding the food waste taken into consideration in the current study are given in Table 3.

**Table 3.** Overview of the food waste considered for each stage of the food supply chain and packaging film type as a percentage of the total agricultural production of tomatoes.

| Food Supply Chain Stage | Food Waste (% of Total Production) | | |
|---|---|---|---|
| | PF | EF and CF | Origin |
| Agricultural production | 23% | 23% | [7,62] |
| Storage and packaging | 4.5% | 4.5% | |
| Distribution and retail | 1.5% | 1.0% | PFW calculated based |
| Consumption | 21% | 13% | on Equation (2) |

*2.4. LCA Methodology*

In this section, the LCA methodology applied in the current study is explained for each of the LCA phases.

2.4.1. Goal and Scope Definition

The current research aims to examine the overall environmental performance of novel food packaging films enriched with antioxidant compounds which are found to extend the shelf life of fresh vegetables. In particular, two new packaging films were studied for the packaging of fresh tomatoes and compared to a conventional packaging PP film.

The extension of the shelf life of tomatoes through the use of such packaging films is expected to lead to reduced food spoilage, and food and related packaging waste at the retail and end-consumer stage, as well as an equal amount of avoided agricultural production and packaging. This in turn will have an effect on the whole supply chain, i.e., reduced transportation, storage, and refrigeration needs related to the avoided food production. To properly capture this effect of the novel packaging films, it was considered necessary to include in the assessment the life cycle of the wasted tomatoes in addition to the life cycle of the packaging material itself. This is in line with the JRC guidelines on the LCA of plastic products in the case of comparing packaging alternatives involving the generation of different amounts of food waste [53].

The adopted effect-oriented modeling approach is more in line with the consequential LCA, which aims to examine the consequences of a product beyond its life cycle by expanding its own system boundaries to include the life cycle of the affected systems as well [64–66].

The LCA was performed based on the ISO 14040 and 14044 protocols [67,68], considering also the methodological framework on the Product Environmental Footprint (PEF) recommended by the European Commission [69] and the detailed rules of the JRC for conducting LCA studies for plastic products [53].

■　　*System boundaries*

The selected system boundaries follow a cradle-to-grave approach, from raw material extraction to consumption and end-of-life treatment. A supply chain logic was applied to include all stages of the individual life cycles of the packaging material and tomatoes.

In particular, the system boundaries of our analysis start from the manufacturing of the packaging material for the tomatoes and the agricultural production of tomatoes, including their processing, continue with the packaging process where the tomatoes are packaged, their distribution to the retail stage, up to their consumption by the end user, and the disposal of food and packaging waste. The system boundaries also include all of the necessary transportation for transferring the raw materials and the intermediate products to the various production stages, the transportation of the final products to the retail stage and the consumer, and the transportation of the waste generated throughout the system stages for their treatment.

Three scenarios are formulated based on the different packaging film alternatives used for the packaging of tomatoes, as presented next:

- Scenario a: The reference scenario using plain packaging film PP film (PF);
- Scenario b: The alternative scenario using a PP film with incorporated tomato leaf-stem extract and Flavomix through extrusion (EF);
- Scenario c: The alternative scenario using a PP film coated with zein nanofibers containing the bioactive compounds of Scenario b through electrospinning (CF).

The difference in the system stages included in the boundaries of the scenarios lies in the addition of a stage for the extract production of the novel packaging materials and its transportation to the plastic film manufacturing plant (Scenario b and c), while all of the other stages remain the same across the different scenarios. The system boundaries of the current LCA study are presented in Figure 1.

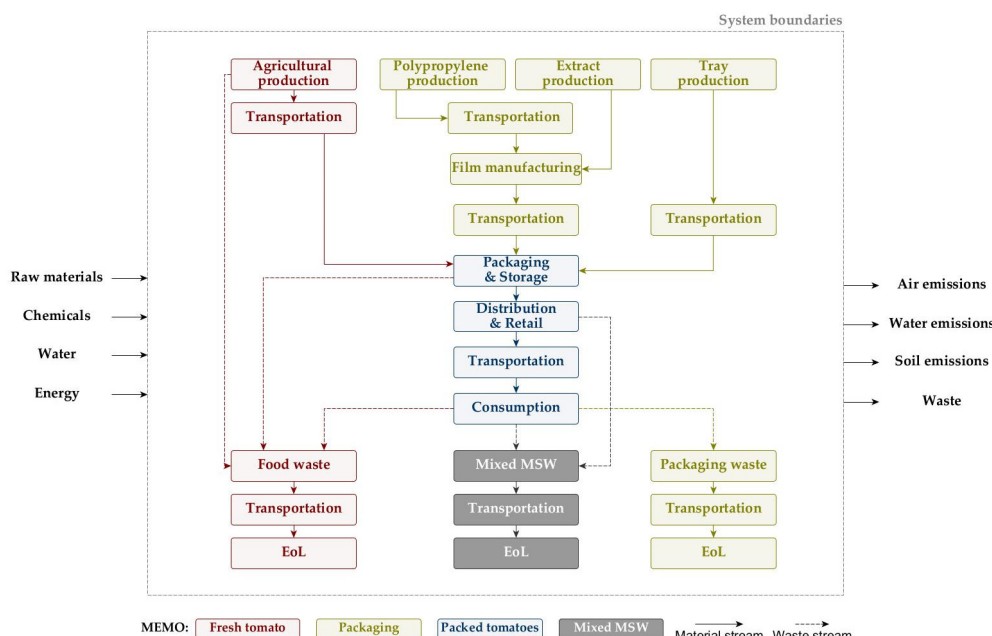

**Figure 1.** System boundaries of the LCA study on the alternative packaging films for tomato packaging.

- *Functional unit and reference flow*

The functional unit is set to the consumption of 1 kg of tomatoes by the end consumer, within the shelf life of tomatoes. Reference flows are estimated for the amount of produced tomatoes needed to fulfill the functional unit, including the amount of tomatoes that is wasted. Based on this, the amount of packaging material needed for meeting the functional unit is also estimated. The reference flows are calculated based on the percentages of food waste across the supply chain of tomatoes (Table 3) and are presented in Table 4.

**Table 4.** Reference mass flows for tomatoes across the supply chain under the different scenarios.

| Food Supply Chain Stage | Scenario a | | Scenario b and c | |
|---|---|---|---|---|
| | Amount | Calculation Formula | Amount | Calculation Formula |
| Total agricultural production [I] | 1.98 kg | Total [I] | 1.70 kg | Total [I] |
| Agricultural production minus FW [II] | 1.52 kg | [II] = [I] − 23% * [I] | 1.31 kg | [II] = [I] − 23% * [I] |
| Packaging and storage minus FW [III] | 1.45 kg | [III] = [II] − 4.5% * [I] | 1.24 kg | [III] = [II] − 4.5% * [I] |
| Distribution and retail minus FW [IV] | 1.42 kg | [IV] = [III] − 1.5% * [I] | 1.22 kg | [IV] = [III] − 1% * [I] |
| Consumption minus FW [V] | 1 kg | [V] = [IV] − 21% * [I] | 1 kg | [V] = [IV] − 13% * [I] |

### 2.4.2. Life Cycle Inventory

The foreground system refers to the production of the novel packaging films and includes input related to their effect on extending the shelf life of tomatoes. The inventory data for the foreground system were supplied by the authors of the Laboratory of Process Analysis and Design, School of Chemical Engineering, National Technical University of Athens team who carried out the lab experiments. All of the other system processes constitute the background system for which existing databases and the literature were used

as the input for their modelling. The detailed inventories for all of the scenarios studied can be found in Tables S3–S5 of the Supplementary Materials. The main LCI database used in the current study is the Ecoinvent version 3.9 [70].

The geographical context was defined to refer to the average situation in Europe in order for the results of the study to be easily applicable to other similar cases in Europe. Therefore, the relevant information was sourced from the available databases referring to the European region (e.g., average European grid electricity mix, production processes for the background systems, waste management options). On the other hand, the stages of tomato cultivation, storage, and transportation were adapted to the Mediterranean context with respect to the ambient temperature conditions (i.e., cultivation in unheated greenhouse, cooling needs for tomato preservation). With respect to electricity, medium-voltage electricity was chosen for the industrial users, while low-voltage electricity was chosen for the retail and the end consumer. Following, the modeling approach considered for each of the system stages is presented.

- *Agricultural production*

The agricultural production of tomatoes was modeled according to the Ecoinvent process "tomato production, fresh grade, in unheated greenhouse", which is based on data provided from farms in Spain. The dataset considers the inputs that are needed for the production of tomatoes intended for fresh consumption as well as the respective emissions, from soil cultivation up to the harvest of tomatoes and their transport to the farm gate.

- *Production of packaging material*

The packaging material consists of a packaging film and a paperboard tray, with the type of the former being differentiated among the scenarios examined. The type and amounts of materials included in the packaging of each scenario are presented in Table 5.

**Table 5.** Type and amount of components needed for producing one reference package of fresh tomatoes.

| Packaging Component | Scenario a | Scenario b | Scenario c |
|---|---|---|---|
| Film | PP: 0.005 kg | • PP: 0.005 kg<br>• Tomato extract +<br>Flavomix: $5.56 \times 10^{-5}$ kg<br>• Oil: $1.11 \times 10^{-4}$ kg | • PP: 0.005 kg<br>• Tomato extract +<br>Flavomix: $5.68 \times 10^{-5}$ kg<br>• Zein: $5.68 \times 10^{-4}$ kg |
| Tray | Paperboard: 0.021 kg | Paperboard: 0.021 kg | Paperboard: 0.021 kg |

For the production of the plain packaging PP film that is used in all three packaging types, the dataset "Oriented polypropylene film E" was selected from the Industry 2.0 database. The dataset includes all processes, inputs, and emissions from the production of oriented polypropylene film, including the production of PP resin, transport of the resin to the converter, the conversion process itself, and packaging of the finished product for onward dispatch.

In this study natural antimicrobial and antioxidant substances were both loaded in zein nanofibers and used as a coating for conventional plastic films and were also directly incorporated in the common film production for the creation of two different active packaging materials. For the production of the active packaging materials (scenario b and c), a series of processes is considered from the collection of tomato waste in the field up to the creation of the final products. First of all, the agricultural waste (tomato leaves and stems) is transported from the field for drying. Drum drying is used for the dehydration of the tomato waste, leading to a 74% reduction in the moisture content. The next step is the milling and sieving of the waste. This stage helps to increase the extraction yield [71], leading to a powder form of a small diameter ($\leq$400 µm). Subsequently, the waste is subjected to ultrasound-assisted extraction (UAE) with ethanol-acetic acid as the solvent system in a 95/5 volume ratio and a solvent/powder weight ratio of 20/1, achieving an encapsulation efficiency of 25%. The extraction mixture is then filtered for the separation of the extract from the rest of the waste. The removal of the solvent from the extract takes

place in a rotary evaporator. Depending on the application that follows, either complete or partial evaporation of the solvent occurs. The removed solvent is condensed using cooling water in order to be reused for extractions. The two final processes for the production of the active packaging films are (a) extrusion and (b) extrusion, followed by electrospinning.

In the first process, a complete solvent evaporation is undertaken and the dry extract together with Flavomix are incorporated into oil (oil/active agents' ratio by weight: 2/1). The mixture is finally inserted into the plastic production line along with PP granules. In both cases, the bioactive content of the final films is 1% *w/w*. All process flow data in this study are based on either simulation data (AspenPlus) or laboratory experimental data that were scaled up based on the literature and Gabi databases (Gabi prof). In the second process, a partial evaporation of the solvent takes place and the concentrated extract is mixed with zein flakes (a prolamin from corn) and Flavomix during stirring. Zein acts as the matrix for the encapsulation of the bioactive compounds (zein/active agents' ratio by weight: 10/1). The mixture is subsequently used as the feed solution in the electrospinning process. At the collector of the electrospinning system, a plain PP film is placed and zein nanofibers containing the tomato extract-Flavomix mixture are formed on its surface as an extra layer. The flow diagram of the processes is presented in Figure 2.

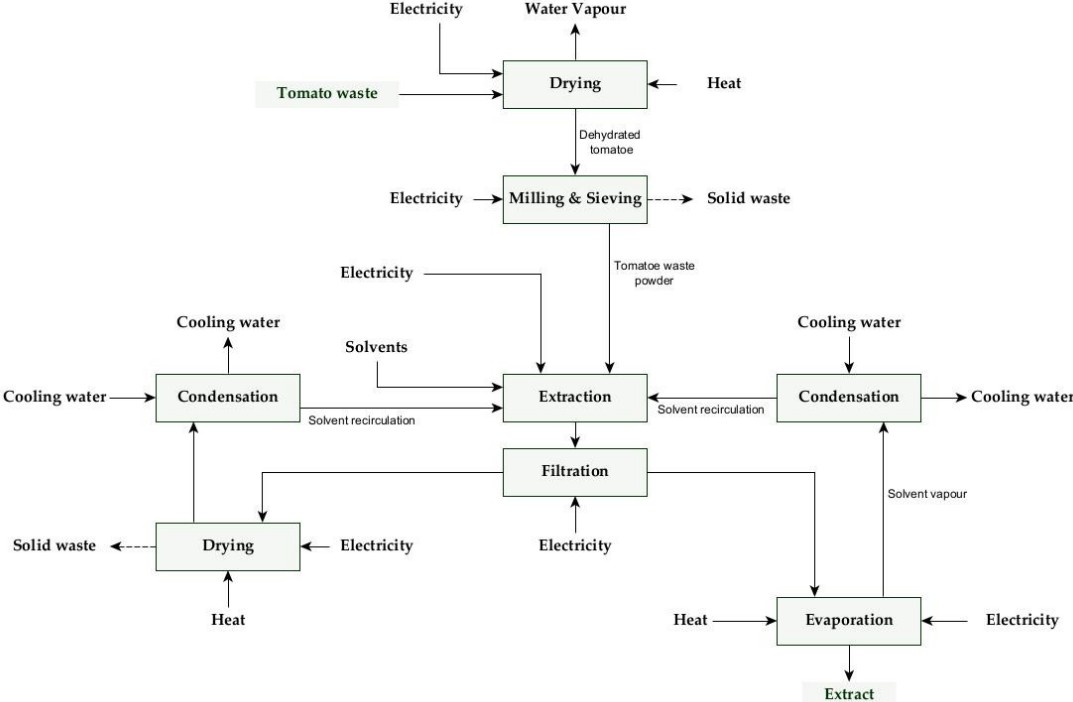

**Figure 2.** Flow diagram of processes for the production of bioactive compounds from tomato waste.

For the production of the paperboard tray, an aggregated dataset was selected from the Ecoinvent database, including the production of white-lined chipboard, transport to converting sites, and converting (printing, cutting, creasing) into blank boxes. The tray is predominantly made of waste paper and paperboard.

■   *Tomato packaging*

According to Frankowska et al. [72], fresh tomatoes undergo some basic cleaning at packaging houses, followed by sorting, packing, and short-term storage. This process was modelled based on the energy and water consumption foreseen in [73].

■　*Storage at retail*

Information concerning the retail stage was sourced directly from a large supermarket chain activated in Greece through personal interviews [74]. Based on the information provided, tomatoes are stored at room temperature (~15–18 °C) next to the refrigerator area of the grocery store.

■　*Household storage*

Following the assumptions of Brown et al. [75], a class A+ refrigerator of 150 L net fridge volume with an average annual consumption of 268 kWh was considered for household storage. The electricity consumption for the refrigeration of tomatoes was estimated based on the total energy consumption of the fridge, its capacity, storage duration, and the specific volume of the product. A gradually reducing volume of the product was assumed based on a constant daily rate of consumption.

■　*End-of-Life treatment*

Throughout the stages of the supply chain of packed fresh tomatoes different types of waste are generated which need to be treated. Information on the adopted waste treatment processes per type of waste and stage is based on Eurostat data [76,77] and relevant published reports [78,79]. In particular, during the Agricultural production stage, spoiled and inappropriate for consumption tomatoes are treated mainly through composting and anaerobic digestion [76,78]. From the extraction and the tomato packaging processes, biowaste is also generated which undergoes the same treatment as above [76]. Waste generated during the Distribution and Retail stage includes tomatoes packed in the tray and wrapped with the plastic film which were considered as mixed municipal solid waste. In this study, mixed municipal solid waste is either incinerated with energy recovery or landfilled [77]. At the Consumption stage, different capture rates were considered for the separate collection of each waste type (tomatoes biowaste, paper tray, plastic film) based on the European Commission [79]. The remaining non-separately collected waste at the consumer level is treated as mixed municipal solid waste as previously mentioned. The respective waste treatment datasets were sourced from Ecoinvent. Table 6 summarizes the waste treatment methods per type of waste and stage of the food supply chain.

**Table 6.** Waste treatment methods per type of waste and stage of the supply chain.

| Stage | Waste Type | Waste Treatment Processes | Ref. |
|---|---|---|---|
| Agricultural production | Tomato Biowaste | • 94.32% Recycling (53% composting; 47% anaerobic digestion [78])<br>• 3.85% Incineration<br>• 1.83% Landfilling | EU-27, 2020 [76] |
| Packaging | Tomato Biowaste | • 94.32% Recycling (53% composting; 47% anaerobic digestion [78])<br>• 3.85% Incineration<br>• 1.83% Landfilling | EU-27, 2020 [76] |
| Distribution and Retail | Mixed municipal solid waste | • 50.5% Incineration with energy recovery<br>• 49.5% Landfilling | EU-27, 2019 [77] |

**Table 6.** *Cont.*

| Stage | Waste Type | Waste Treatment Processes | Ref. |
|---|---|---|---|
| Consumption | Source separated:<br>• Plastic<br>• Paper<br>• Biowaste | • 36% Paper capture rate<br>• 11% Plastic capture rate<br>• 15.5% Biowaste capture rate | EU-28, 2020 [79] |
| | Source separated Plastic | • 71.3% Recycling<br>• 23.2% Incineration with energy recovery<br>• 5.5% Landfilling | EU-27, 2020 [76] |
| | Source separated Paper | • 98.33% Recycling<br>• 1.51% Incineration with energy recovery<br>• 0.16% Landfilling | EU-27, 2020 [76] |
| | Source separated Biowaste | • 94.32% Recycling (53% composting; 47% anaerobic digestion [78])<br>• 3.85% Incineration<br>• 1.83% Landfilling | EU-27, 2020 [76] |
| | Remaining waste (non-separately separated) | • 50.5% Incineration with energy recovery<br>• 49.5% Landfilling | EU-27, 2019 [77] |
| Antioxidants extraction from tomato waste | Tomato Biowaste | • 94.32% Recycling (53% composting; 47% anaerobic digestion [78])<br>• 3.85% Incineration<br>• 1.83% Landfilling | EU-27, 2020 [76] |

■ *Transportation*

Transportation is considered in each life cycle stage, including ambient and refrigerated trucks. It is assumed that transportation takes place mostly at a regional (NUTS2) level, where the tomato production, the packaging material production, the tomato packaging facility, as well as the retail stores, the consumer, and the waste treatment facilities are located within the same region. The only exception is considered for the production of extracts, which is assumed to be located in the same facility as the film production.

In particular, a refrigerated truck was considered for transport from the packaging facility to retail, as it is compulsory in all super market chains to deliver fruits and vegetables by refrigerated truck [74]. For the transportation of the product from the retail to the consumer household, transport by car was considered at a rate of 70% (the remaining 30% was assumed to be transported on foot), using an allocation factor of 0.012 based on the tomatoe's specific volume, according to the European Commission [53] recommendations. Table 7 summarizes the distances and type of vehicles assumed for road transport.

**Table 7.** Distances and type of vehicles assumed for road transport between the life cycle stages of the study.

| Life Cycle Stage | Distance (km) | Truck Type | Reference |
|---|---|---|---|
| Field to packaging facility | 50 | Truck | Assumption based on [80] |
| Tomato waste to extract production | 150 | Truck | Authors' assumption |
| Packaging materials to the packaging facility | | | |
| To retail | 100 | Lorry with refrigeration | Authors' assumption |
| To consumer | 5 | Passenger car | [53] |
| To waste treatment facilities | | | |
| *AD, incineration, landfill* | 100 | Truck | [53] |
| *Compost* | 30 | | |

2.4.3. Life Cycle Impact Assessment

The modeling and evaluation of the different scenarios were performed using SimaPro 9.4. The Environmental Footprint (EF) Version 3.0 method [81–83] was used in this study for conducting the life cycle impact assessment (LCIA), as recommended by the European Commission [81]. Within the EF method, the normalization factors are expressed per capita.

The EF method comprises 16 midpoint impact categories, from which 6 were identified to be the most relevant ones based on the results of the analysis and were selected to be presented here. It was decided that only the impact categories that cumulatively contribute to the total weighted environmental impact by 82% would be considered. The selected impact categories are by order of magnitude: Ecotoxicity, freshwater (EcF), Climate change (CC), Water use (WU), Particulate matter (PM), Resource use, fossils (RUf), and Resource use, minerals and metals (RUmm).

**3. Results**

In this section, the results of the current study on the technical and environmental performance of novel films for packaging fresh tomatoes using LCA are presented.

With respect to the properties of the novel films, both films showed sufficient antioxidant capacity ranging from 12 to 20% inhibition of DPPH free radicals. Specifically, the extruded film showed the lowest % inhibition due to the difficult diffusion of antioxidant compounds from the PP matrix, whereas the coated films showed the highest one due to the direct release of the bioactive compounds from the surficial coating. Regarding the antimicrobial activity, both films showed sufficient antimicrobial activity towards Gram negative (*E. coli*) and very low against Gram positive (*Staphylococcus aureus*) bacteria, as shown in Table 8.

**Table 8.** Antimicrobial activity (R) of PP films containing bioactive compounds.

| Type of Film | *Staphylococcus aureus* (Log Cells/cm$^2$) | *Escherichia coli* (Log Cells/cm$^2$) |
| --- | --- | --- |
| Control (PF) | 0 | 0 |
| Extruded (EF) | 1.3 | 2.1 |
| Coated (CF) | 1.6 | 2.9 |

According to the standard ISO 22196:2011, an antibacterial product is determined to have antibacterial effectiveness when the antibacterial activity (R) is more than 2 Log cells/cm$^2$.

The results indicated the superiority of the active materials produced and the achievement of an extended shelf life compared to the conventional film. The plain PP film (Film a) acted as a typical package leading to an expected 4–5 day shelf life until the yeast and molds content exceeded the permissible limits for consumption. In contrast, the EF (Film b) and CF (Film c) with the incorporated bioactive agents offered a shelf-life extension of about 40–75% (2–3 days), depending on the storage temperature. This is due to the gradual release of the encapsulated bioactive compounds. The analytical results of the achieved shelf life for the tomato fruits under the different scenarios and temperature conditions are presented in Table S2 of the Supplementary Materials.

In Figure 3, a comparative overview of the total environmental impacts for the three studied scenarios is presented for the selected impact categories. The results are presented as percentages of Scenario a, which is the reference case and is set as 100%. As is shown and taking into consideration the whole supply chain of the studied product, the application of novel films (Scenarios b and c) results in an improved environmental performance in all impact categories in comparison with the plain PP film (Scenario a). In more detail, both Scenario b and Scenario c show a decrease of approximately 14% for each impact category in comparison with Scenario a. Scenario c had a slightly better environmental performance on average in comparison with Scenario b by 0.2%, for the selected impact categories.

Information on the absolute values of the total environmental impacts per selected impact category for each studied scenario is presented in Table 9.

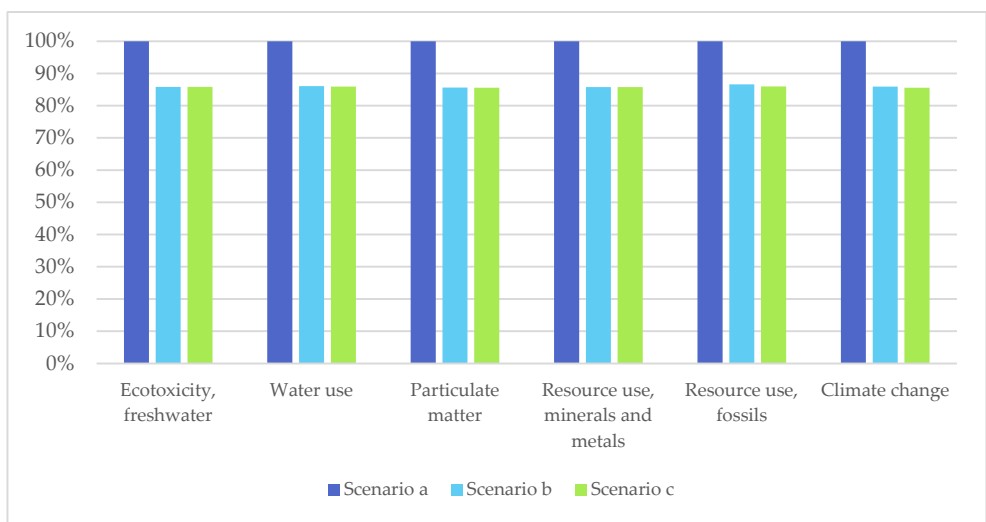

**Figure 3.** Relative comparison of environmental impacts of the three scenarios studied.

**Table 9.** Total characterized impacts of the three scenarios for the selected impact categories.

| Impact Category | Unit | Scenario a | Scenario b | Scenario c |
|---|---|---|---|---|
| Ecotoxicity, freshwater | CTUe | 95.675028 | 82.107498 | 82.105956 |
| Water use | $m^3$ depriv. | 4.4157863 | 3.8000763 | 3.7943258 |
| Particulate matter | disease inc. | $1.74 \times 10^{-7}$ | $1.49 \times 10^{-7}$ | $1.49 \times 10^{-7}$ |
| Resource use, minerals and metals | kg Sb eq | $1.44 \times 10^{-5}$ | $1.23 \times 10^{-5}$ | $1.23 \times 10^{-5}$ |
| Resource use, fossils | MJ | 14.55842 | 12.607114 | 12.514904 |
| Climate change | kg $CO_2$ eq | 1.2764677 | 1.0964276 | 1.0920995 |

In Figure 4, the total normalized environmental impacts per supply chain stage including the film production are illustrated for each of the studied scenarios. For all of the three scenarios, the environmental burdens caused by the agricultural production of tomatoes are the main contributors of the resulting impacts with a contribution of approximately 90% of the total impact for each scenario. Moreover, it is observed that Scenarios b and c present an improved environmental performance in comparison with the baseline scenario (Scenario a) for all stages of the supply chain, excluding film production. When focusing only on film production enriched with antimicrobial and antioxidant compounds, it is evident that the novel films production line shows increased environmental impacts compared to the plain PP film. More specifically, the production of the EF film (Scenario b) shows an increased impact of 31% in comparison with the plain PP film (Scenario a), while CF film (Scenario c) shows an increased impact of 18% in comparison with the plain PP film (Scenario a). The increase in the environmental impacts for both scenarios is attributed to the additional processes as well as auxiliary materials needed for the extraction of the anti-compounds. Regarding the largest environmental burdens of Scenario b with respect to Scenario c just for the film production stage, these are mainly due to the different auxiliaries used in each scenario. Additionally, Scenario c has a slightly lower demand on energy, which also contributes to its overall better performance in comparison with Scenario b.

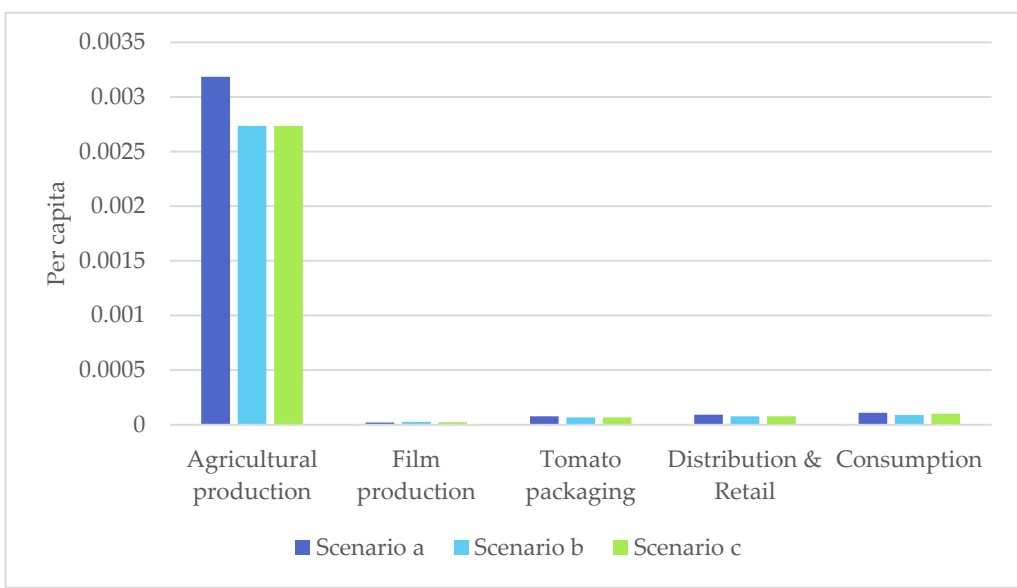

**Figure 4.** Total normalized environmental impact per scenario for the supply chain stages including film production.

Taking a closer look on the decreased environmental burdens of Scenarios b and c, these are attributed to the extension of the product's shelf life that is achieved using active packaging. In more detail, the extended shelf life decreases waste generation, leading to decreased waste management treatment activities across all of the stages of the supply chain. Additionally, since less tomato is wasted, the production of fresh tomatoes and packaging materials also decreases. This results in decreased transportation needs both for the raw materials and the waste produced. Regarding film production, the novel materials production line shows a worsened overall impact compared to the production line of the baseline packaging material. This fact is attributed to the additional environmental burdens caused by the extraction processes needed to isolate and integrate the antimicrobial compounds, as well as the treatment of the process-specific waste streams (i.e., biowaste and spent solvent).

In Figure 5, the allocation of the impacts of the different groups of activities to the selected impact categories is presented for Scenarios a and c. Six groups were formulated, namely Material, Process, Waste treatment, Agricultural production, Energy, and Transportation. The contribution of the groups to the total impact score follows a similar trend among the selected impact categories in Scenarios a and c.

In most impact categories, the Agricultural production group has the highest share of impact, followed by the Transportation and Waste treatment groups. Focusing on each impact category, Agricultural production is the main contributor in the Climate change impact category, accounting for the 59.6% of the total impact, followed by the Waste treatment (17.7%)—mainly due to sanitary landfill of MSW—and Transportation (15.1%) groups. For the impact category of Resource use, fossils, the Agricultural production group has the highest contribution (66.4%) to the total impact, followed by the Transportation (18%), Process (7.5%), and Energy (6%) groups. In the impact categories of Particulate matter and Resource use, minerals and metals, Agricultural production contributes to almost 88% in both categories, which is directly linked to fertilizer use, followed by the Transportation group contributing approximately 7.3% and 9.1%, respectively, per impact category. The highest share of impacts on the Water use category is from Agricultural production (97.5%), which is due to the irrigation needed for tomato production. The impacts on Ecotoxicity, freshwater from Agricultural production (90.8%) mainly derive from fertilizer use and plastic construction materials used for the greenhouse construction, while the impacts from the Waste treatment group (4.7%) are caused by industrial composting.

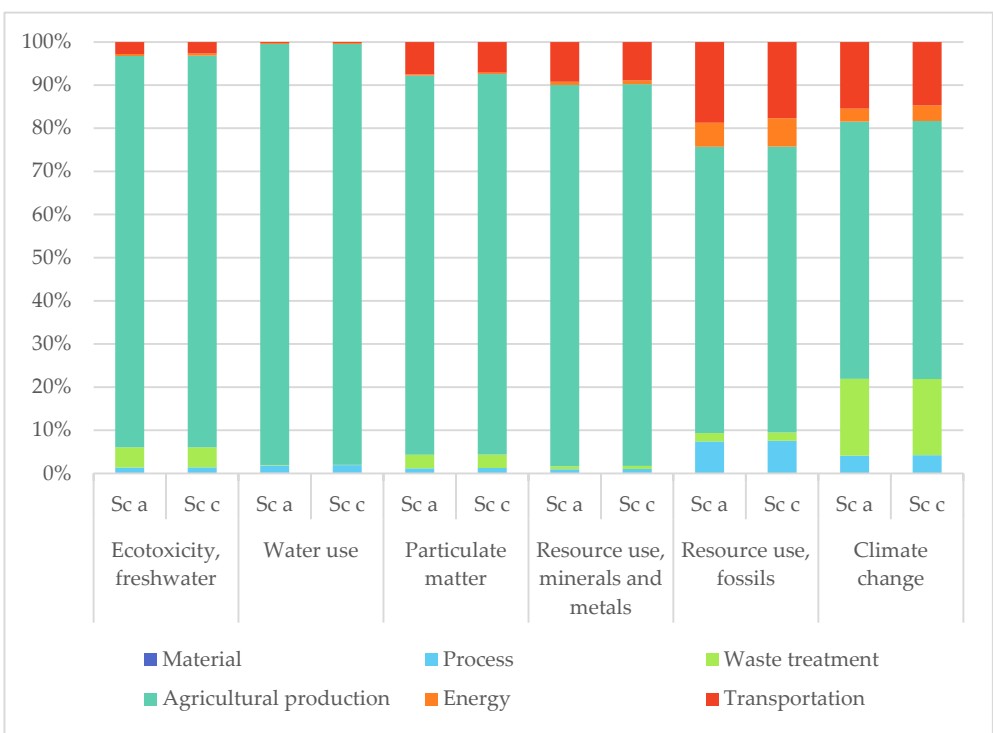

**Figure 5.** Contribution of group of categories per impact category for Scenario a (Sc a) and Scenario c (Sc c).

Focusing on the aforementioned groups, for the Process group, the film and tray production processes have higher contributions, while in the Transportation group most impacts arise from the refrigerated transportation to retail and passenger car transportation. Additionally, most of the impacts in the Energy group are linked to the electricity consumption for film production and the operation of the packaging facility. Scenarios b and c follow the same pattern regarding the total impact contribution per group. Relative changes are observed in the Process and Energy groups, where Scenario c results in a lower contribution to the Resource use, fossil (−4.8%) and Climate change (−4.4%) impact categories in comparison with Scenario b due to the improved performance of the electrospinning process.

## 4. Discussion

In this section, a discussion is made on the results of the current study as well as of similar LCA studies based on the considered parameters. Nevertheless, a direct comparison was not feasible as several modelling parameters (e.g., system boundaries, shelf life, supply chain, food production and food waste reduction, etc.) were different. As the selected impact assessment methods (LCIA) varied across the studies, a focus is given in those impact categories that are the same or similar to the ones examined in our study.

In brief, in this study the life cycle environmental performance of active packaging solutions (EF and CF) extending tomato shelf life was investigated against conventional packaging film (plain PP). The effect of shelf-life extension on the reduction of food waste and food production was taken into account. The system boundaries included the whole supply chain of tomato and its packaging, up to the consumer and waste treatment stages. It was estimated that for an increase in shelf life by 1.6 times (i.e., an increase of 60%, Table 2) measured with the EF and CF films, a decrease of 29% in food waste throughout the supply chain and a subsequent reduction of 14% in food production (Table 4) could be achieved, based on the SLR formula. It is concluded that the percentage reduction in the total impacts (14%, Figure 3) is mostly attributed to the reduction in food production, considering that

the impacts related to the agricultural production phase dominate the total impact score with a contribution of 63–93% in the respective impact categories (Figure 5).

Gutierez et al. [46] studied the effect of a MAP film for extending the shelf life of cheesecake compared to a conventional LDPE film. The MAP film was found to increase the product shelf life by four times compared to the LDPE film, which was translated into a proportionate reduction in food loss based on the use of an economic formula. The latter assumes that the extension of shelf life results in higher sales and less distribution needs, with the amount of food production remaining the same. The system boundaries are the same as the ones adopted in our study, with the exception that the consumer stage is excluded. The results showed a decrease in environmental impacts in the case of MAP ranging from 29 to 34% across the relevant (to our study) impact categories studied. The decrease was mainly attributed to the savings in food loss and to the distribution process. Overall, Gutierez et al. [46] found that the production of food had the highest contribution (up to 90%) in both solutions examined, while the distribution and waste treatment phases had a considerably lower share. Therefore, the reduction in the impact in these stages is not so obvious in the overall impact results.

In the same direction, Settier-Ramirez et al. [13], who studied the environmental impacts of pastry cream packaging film coated with natural antimicrobial compounds versus a conventional PE packaging film, found that the environmental load of the coating production is offset due to the extension of the product shelf life. In particular, an extension of shelf life by 4.3 times (from 3 to 13 days) was achieved, which was correlated to a change in food waste generation at the manufacturing and retail stages. In particular, for the manufacturing stage an empirical estimation was made. For the retail stage, the ex-ante probability of the pastry cream not being sold before shelf life expiration was calculated based on a continuous linear formula, assuming that the product had a constant probability of being sold each day it was displayed on the shelf. This resulted in the estimation of a reduction in the total waste generated throughout the supply chain by 87%, as well as in a reduction in pastry cream production of 42%. It is noted that the examined system boundaries include the whole life cycle of the pastry cream-package system, including the end user stage. The results of the life cycle impact assessment show a reduction in total impacts in the range of 35–43% with the use of the active packaging for the relevant (to our study) impact categories. The contribution analysis also shows that the production of pastry cream (including the wasted pastry cream) is responsible for the majority of the impacts, accounting for 84–100% of the total environmental burdens. Again, it can be noticed that the reduction in the pastry cream production achieved (42%) with the use of active packaging is very close to the observed decrease in environmental impacts, as this stage has the highest contribution of impacts (35–43%). This is also the case in our study, as mentioned previously in this section.

The study of Zhang et al. [45] focuses on the comparative life cycle assessment of the use of essential oil component-enabled (active) packaging for extending the shelf life of fresh beef against conventional packaging (MAP). In particular, three active MAP solutions were examined considering a food loss reduction at the retail stage by maximum 1.8% based on the literature, while an equivalent reduction in beef production was also assumed. It is noted that the system boundaries include all relevant stages of the food supply chain including packaging, except for the consumer and the waste treatment stages. As most of the studies reviewed, the authors concluded that the novel packaging solutions resulted in an improved eco-profile across the impact categories evaluated. Specifically, a reduction of 1.7% and 1.4% was achieved in the global warming and the cumulative energy demand categories, respectively, with the use of the best performing packaging solution. Considering that the impacts of the food production stage (cattle fattening) dominate the total impact score (82–96% share), it is reasonable to assume that the percentage reduction in food production is mostly responsible for the reduction in the total impact score.

Manfredi et al. [44] aimed to assess the environmental impacts arising from the application of an antimicrobial coating onto the packaging of a fresh milk product, while

also taking into account the reduction in milk waste due to the extension of its shelf life. In particular, an extension of shelf life by 3.5 times (from 2 to 9 days) was considered based on the literature. Then, a reduction in milk waste at the consumption stage by 33% (from 7% to 4.7%) was assumed, resulting in a 2.6% reduction in milk production. The system boundaries consider the whole life cycle of the milk-package system, from raw milk production to potential milk waste and the packaging's life cycle. The authors found that the potential reduction in milk waste using the coating generates higher environmental benefits than the burdens caused by the coating's life cycle due to milk saving. Specifically, the average reduction in impacts across the examined categories is about 2%. It is also noticed here that the life cycle of milk production (including wasted milk) constitutes the main cause of impacts, representing 70–100% of the total environmental burdens. As a result, we observe again here that when the stage of food production is included in the analysis and its contribution is high in the overall impacts, the resulted reduction in the total impact score is mainly attributed to the assumed reduction in food production.

## 5. Conclusions

Food and packaging waste is generated throughout the entire food supply chain of fresh vegetables and fruits from primary production to consumers. To minimize both types of waste, new packaging materials incorporating active ingredients in polymeric matrices can play a fundamental role as they can extend the shelf life of the products. This study aimed to evaluate the environmental benefits of two novel food packaging materials by integrating the extension in the shelf-life extension of fresh packed tomatoes. The antioxidant and antimicrobial activity of both films led to a 3 day extension of the shelf-life by inhibiting and postponing the development of yeasts and molds and total viable bacterial count. On the whole, the results showed that both new packaging films performed better than the conventional polypropylene film, when taking into account all stages of the food supply chain. Based on what was previously presented, it is observed that Scenarios b (EF) and c (CF) are found to be of an improved environmental performance in comparison with the baseline scenario (Scenario a—PF) for all stages of the supply chain, excluding film production. When focusing only on the novel film production, it is evident that the environmental impacts increased in comparison with the conventional PF film.

Interpreting the results of the current study along with the results of previous similar studies, it can be said that the contribution of the environmental burdens due to the production of novel packaging solutions is minimized when considering the entire food supply chain, since the environmental burdens arising from the primary production are predominant in the total environmental impacts. Furthermore, encapsulating the avoided food waste and food production as a result of improved packaging alternatives constitutes a challenge in terms of estimating and quantifying the food waste within the several stages of the food supply chain. Therefore, future studies could focus on providing a solid base with the acquisition of empirical data on the correlation between the extension of shelf life and the reduction in food waste. Finally, to support the uptake and commercialization of the examined novel packaging materials, it is necessary to also provide information on the economic viability of the implementation of such solutions on a large scale.

**Supplementary Materials:** The following supporting information can be downloaded at: https://www.mdpi.com/article/10.3390/su15107838/s1, Table S1: Specifications and maximum levels of acceptance of the compact Dry discs used in the experiment; Table S2: Results of shelf life study on tomato fruits stored at 5 °C, room temperature and 45 °C and a relative humidity of 80% for 1 week.; Table S3: Life Cycle Inventory for Scenario a.; Table S4: Life Cycle Inventory for Scenario b; Table S5: Life Cycle Inventory for Scenario c.

**Author Contributions:** Conceptualization: C.T., C.P. and K.V.; Methodology: C.T., C.P., K.V., A.K. and P.A.; Software: C.T., A.K. and P.A.; Validation: C.T. and A.K.; Formal analysis: C.T., C.P., K.V., A.K. and P.A.; Laboratory analysis and research: M.P. and S.P.; Resources: P.A., A.K. and M.P.; Extraction processes upscaling: C.B., M.P. and S.P.; Supervision: C.P., K.V., S.P. and M.K. All authors have read and agreed to the published version of the manuscript.

**Funding:** This research has been co-financed by the ERDF of EU and Greek national funds through the Operational Program Competitiveness, Entrepreneurship and Innovation, under the special actions "Aquaculture"—"Industrial Materials"—"Open Innovation in Culture" (project code: T6ΥΒΠ-00220, MIS 5048495).

**Institutional Review Board Statement:** Not applicable.

**Informed Consent Statement:** Not applicable.

**Data Availability Statement:** Not applicable.

**Acknowledgments:** We sincerely thank AB Vassilopoulos super market and, in particular, Stefanos Gkinosatis for providing technical details and description on the storage and transportation of the fresh tomatoes across the supply chain.

**Conflicts of Interest:** The authors declare no conflict of interest. The funders had no role in the design of the study; in the collection, analyses, or interpretation of data; in the writing of the manuscript; or in the decision to publish the results.

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
