# Peer review of "Investigating the Environmental Benefits of Novel Films for the Packaging of Fresh Tomatoes Enriched with Antimicrobial and Antioxidant Compounds through Life Cycle Assessment"

_sustainability, doi:10.3390/su15107838_

Round 1

Reviewer 1 Report

I am much honored to have been allowed to review the manuscript.

Please use its full term (Life Cycle Assessment) instead of LCA in the title of the article.

The abstract should be more informative by giving real results rather than elastic sentences. Important and main contents should be given. Support the results with some quantitative data.

I suggest that you use new articles such as the following in the initial parts of the introduction: 10.1007/s11694-021-01235-8; 10.1186/s40538-023-00393-9.

Include the novelty of the study in the introduction.

There are several spelling and grammar mistakes in the manuscript. Many loose sentences without providing actual meaning have been found. Read thoroughly and correct them.

The discussion section needs more specific detailed comparative studies. This part is very inadequate. Please compare with similar works after presenting each result. Improve this section carefully.

Conclusion: what is the future of your findings? The conclusion is not insightful, what are your suggestions?

Rewrite the references according to the journal format.

There are several spelling and grammar mistakes in the manuscript. Many loose sentences without providing actual meaning have been found. Read thoroughly and correct them.

Reviewer 2 Report

Dear authors.

Before considering the publication of your MS, I suggest some changes. The results should be in that section. Improve the discussion to clarify and provide a full explanation of your results. Write properly the references in the text. Improve the figure captions, providing the abbreviation meaning. Rewrite the conclusion section, due to repeat the results.

Just minors EN mistakes were identified. 

Reviewer 3 Report

First, I would like to congratulate the authors for the well written article with an interesting topic.

This article compares the performance and LCA of two novel food packaging for fresh tomatoes, with a conventional polypropylene solution.

Some considerations that might improve the quality of the article:

·       Line 183: What is the polymer matrix of the EF films?

·       Materials and Methods: there should be a sub-section describing how the EF and CF films were produced. If they are commercially available, or if there was used a methodology already described in the literature, it should be cited in the text.

Materials and methods are very extensive. Some reformulation could help to clarify some ideas.

Reviewer 4 Report

Dear Authors,

This manuscript is research on “Investigating the environmental benefits of novel films for the packaging of fresh tomatoes enriched with antimicrobial and antioxidant compounds through LCA”.  I would like to say that the research is unacceptable without major revision. comments are attached as a file

Minor editing of English language required

Round 2

Reviewer 1 Report

All corrections are well done.

Reviewer 4 Report

I am pleased to inform accept of revised manuscript.